# A System-Level Exploration of Binary Neural Network Accelerators with Monolithic 3D Based Compute-in-Memory SRAM

**Jeong Hwan Choi [1,†], Young-Ho Gong [2,\*] and Sung Woo Chung [3,\*]**

1    SK Hynix, Gyeonggi-do 13558, Korea; jeonghwan2.choi@sk.com
2    School of Computer and Information Engineering, Kwangwoon University, Seoul 01897, Korea
3    Department of Computer Science and Engineering, Korea University, Seoul 02841, Korea
\*    Correspondence: yhgong@kw.ac.kr (Y.-H.G.); swchung@korea.ac.kr (S.W.C.)
†    This work is done when Jeong Hwan Choi worked at Korea University.

**Abstract:** Binary neural networks (BNNs) are adequate for energy-constrained embedded systems thanks to binarized parameters. Several researchers have proposed the compute-in-memory (CiM) SRAMs for XNOR-and-accumulation computations (XACs) in BNNs by adding additional transistors to the conventional 6T SRAM, which reduce the latency and energy of the data movements. However, due to the additional transistors, the CiM SRAMs suffer from larger area and longer wires than the conventional 6T SRAMs. Meanwhile, monolithic 3D (M3D) integration enables fine-grained 3D integration, reducing the 2D wire length in small functional units. In this paper, we propose a BNN accelerator (*BNN_Accel*), composed of a 9T CiM SRAM (*CiM_SRAM*), input buffer, and global periphery logic, to execute the computations in the binarized convolution layers of BNNs. We also propose *CiM_SRAM* with the subarray-level M3D integration (as well as the transistor-level M3D integration), which reduces the wire latency and energy compared to the 2D planar *CiM_SRAM*. Across the binarized convolution layers, our simulation results show that *BNN_Accel* with the 4-layer *CiM_SRAM* reduces the average execution time and energy by 39.9% and 23.2%, respectively, compared to *BNN_Accel* with the 2D planar *CiM_SRAM*.

**Keywords:** monolithic 3D integration; compute-in-memory; binary neural network; energy efficiency

## 1. Introduction

For the deployment of neural networks, the embedded systems (e.g., mobile device) generally focus on the inference (e.g., face recognition) rather than the training, due to the limited resources and energy budgets of the systems. Nevertheless, even for the inference, convolution neural networks (CNNs) still require a huge volume of high precision (e.g., 32-bit or 64-bit) parameters and time consuming multiply-and-accumulate computations (MACs). Thus, it poses substantial challenges for the deployment of CNNs in the embedded systems. In contrast, binary neural networks (BNNs) [1] reduce the precision of the parameters to a single-bit. In addition, BNNs replace expensive MACs with bitwise XNOR followed by population count (popcount) computations; XNOR followed by popcount computation is called as XNOR-and-accumulation (XAC). Thus, BNNs are known to be suitable for resource- and energy-constrained embedded systems compared to CNNs, by reducing the computational complexity as well as the memory footprint with minimal degradation in accuracy (less than 10% [2]).

Due to the simple computational complexity of XAC, the data movements dominate the latency and energy consumption [3]. When designing BNN hardware with the conventional von Neumann architecture (i.e., processing units and memory subsystems), the BNN hardware suffers from substantial latency and energy costs due to the data movements (e.g., the data movement between CPU and off-chip memory consumes ~100× higher energy

than a floating-point operation itself [4]). On the other hand, the compute-in-memory (CiM) technology significantly reduces the latency and energy of the data movements compared to the von Neumann architecture, by enabling computations in the memory array. Recently, several researchers implemented XAC on *CiM SRAMs* by adding additional transistors to the conventional 6T SRAM for BNN [3,5–7]. However, though such CiM SRAMs reduce the latency and energy of the data movements, the CiM SRAMs with the 2D planar structure suffer from the large cell area overhead (e.g., a 12T CiM SRAM has $2.7\times$ larger cell area than the 6 T SRAM [3]), due to the additional transistors. Furthermore, such CiM SRAM architecture composes many subarrays together to enable parallel computations [7], which results in longer wire overhead between subarrays.

Meanwhile, through-silicon-via (TSV) based 3D (TSV-3D) integration has gained huge attention by reducing overall wire length, which results in latency and energy reductions compared to 2D integration. However, TSV-3D integration is not feasible for fine-grained 3D integration of small functional units (e.g., L1 caches), since the microscale TSVs incur non-negligible area overhead [8]. To overcome this drawback, monolithic 3D (M3D) integration is considered as a promising technology for fine-grained 3D integration, thanks to extremely small (nanoscale) monolithic inter-tier vias (MIVs) [8]. Thus, several studies adopted M3D integration to devise 3D integrated small functional units [8–10]. For example, a 9T CiM SRAM was implemented with the transistor-level M3D integration, where additional transistors are vertically integrated with MIVs, resulting in the same footprint as the conventional 6T SRAM [9]. Furthermore, Gong et al. applied the subarray-level M3D integration (i.e., bitline partitioning (BLP) or wordline partitioning (WLP)) to L1 caches, which reduces the length of routing wires between subarrays by replacing 2D wires with extremely small MIVs [8].

In this paper, to accelerate the computations in the binarized convolution layers of BNNs, we propose an energy-efficient *BNN accelerator* (denoted as *BNN_Accel*) composed of a 9T *CiM SRAM*, input buffer, and global periphery logic (the global periphery logic obtains the binarized result by accumulating the partial popcounts and comparing the total popcount with a threshold). We propose the 9T *CiM SRAM*, which performs XACs, with the subarray-level M3D integration to reduce the length of routing wires between subarrays; we also adopt the transistor-level M3D integration for each cell of the 9T CiM SRAM to reduce the cell area [9]. Furthermore, we reveal the potential benefits of our proposed *BNN_Accel* with a commonly exploited BNN (i.e., binarized VGG-16), in terms of the execution time and energy consumption, compared to *BNN_Accel* with the 2D 9T *CiM SRAM*. To our best knowledge, this is the first study to investigate the system-level impacts of subarray-level M3D integration in the *CiM SRAM* on a *BNN accelerator*.

## 2. Related Work

Several studies proposed the CiM SRAMs which perform XAC or XNOR computations [5–7,9,10]. Liu et al. and Yin et al. proposed the CiM SRAMs for XAC based on analog computing, which exploit a multilevel sense amplifier (MLSA) [6] and flash ADC [7], respectively, to compute the popcount. Though the analog-based CiM SRAMs achieved high throughput, they degraded the network accuracy due to the nonlinear quantization of the MLSA and ADC. In contrast, Agrawal et al. exploited a digital bit-tree adder in their 10T CiM SRAM for the popcount computation, achieving the ideal network accuracy [5]. However, the 10T CiM SRAM still incurred the cell area overhead due to the additional transistors. To reduce the cell area overhead, several studies proposed the M3D CiM SRAMs for the XNOR computation, which placed additional transistors on the upper layer [9,10]. The M3D 9T CiM SRAM had the same footprint as the conventional 6T SRAM [9]. However, the M3D 10T CiM SRAM required additional AND gates and global wires in the bottom layer, causing the subarray area overhead [10]. In addition, the previous studies on the M3D CiM SRAMs did not provide the system-level analysis but focused on the cell characteristics (e.g., current-voltage characteristic). In this paper, we propose a subarray-level M3D integration based 9T CiM SRAM with the digital popcount unit

by adopting the cell design from [9] and provide the system-level analysis of *BNN_Accel*, which is based on the M3D 9T CiM SRAM.

### 3. *CiM_SRAM*-Based *BNN_Accel*

*3.1. CiM_SRAM Subarray*

Figure 1 shows the design and operation procedure of a *CiM_SRAM* subarray; in this paper, *CiM_SRAM* denotes the 9T CiM SRAM which is able to compute XAC operations in memory. The *CiM_SRAM* subarray consists of a decoder, row-wise wire (wordline (WL) and signal B) driver, column-wise wire (bitline (BL), bitline-bar (BLB) and signal L/R) driver, 512-by-128 cell array, and Popcount unit. Note we adopt the cell design of *CiM_SRAM* from [9]; three transistors (NL, NR, and NB) and three wires (signal L, R, and B) are added to the conventional 6T SRAM cell. The operation procedure of the *CiM_SRAM* subarray is as follows:

1. Based on an input row address, the signal B turns on all the NB transistors in the selected row.
2. Based on inputs from the outside of the subarray, each signal L/R turns on/off the transistor NL/NR.
3. As the stored value (Q/QB) in each cell is passed to Cout, the XNOR computation is performed.
4. The XNOR results of all the cells in the selected row are passed to the Popcount unit through BLs.
5. XAC is completed as the Popcount unit computes the popcount.

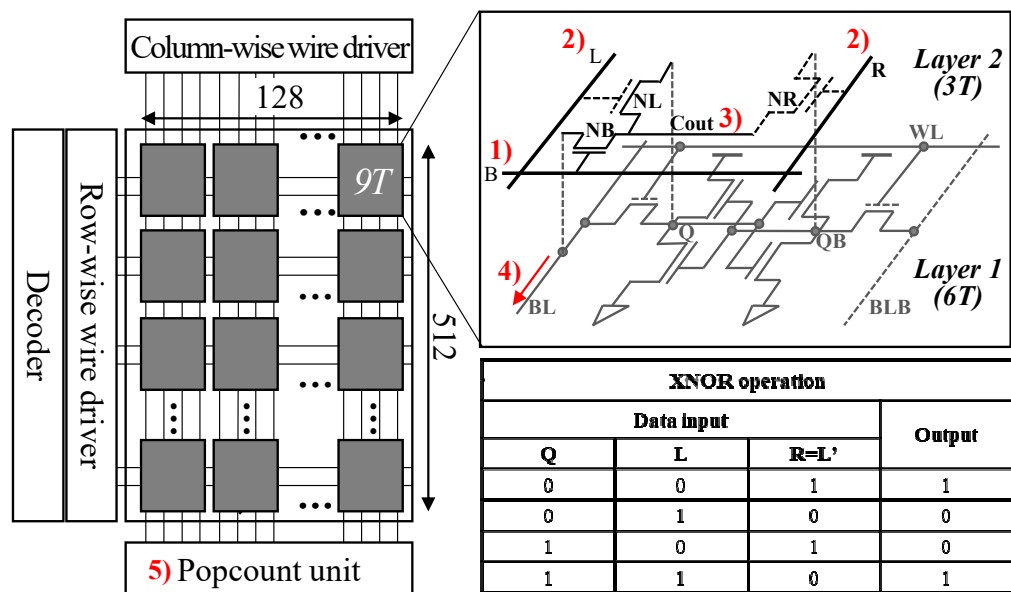

**Figure 1.** Design and operation procedure of a *CiM_SRAM* subarray. Note the design of *CiM_SRAM* subarray is reproduced from [9] and we demonstrate the truth table of XNOR operation in the *CiM_SRAM* design.

Figure 2 presents three different *CiM_SRAM* designs in this paper. Figure 2a shows the 2D planar *CiM_SRAM*, which is denoted as *CiM_SRAM (2D)* [9]. *CiM_SRAM (2D)* causes 1.5× cell area overhead compared to the conventional 6T SRAM, due to three additional transistors. As shown in Figure 2b, *CiM_SRAM (M3D_2L)* [9] is the 2-layer *CiM_SRAM*, adopting the transistor-level M3D integration to *CiM_SRAM (2D)*. Since we vertically interconnect the additional transistors and the conventional 6T SRAM with three MIVs for each cell, *CiM_SRAM (M3D_2L)* achieves the same cell area as the 6T SRAM. Furthermore, by adopting the subarray-level M3D integration (the M3D BLP) to *CiM_SRAM (M3D_2L)*, we propose the 4-layer *CiM_SRAM*, which is denoted as

*CiM_SRAM (M3D_4L).* To design *CiM_SRAM (M3D_4L)* with the M3D BLP, we divide each subarray of *CiM_SRAM (M3D_2L)* to four layers by partitioning the column-wise wires. Thus, *CiM_SRAM (M3D_4L)* has shorter routing wires between subarrays as well as column-wise wires than *CiM_SRAM (M3D_2L).* Note we exploit the BLP, since *CiM_SRAM* has more column-wise wires than row-wise wires. When the row-wise wires consume more latency and energy than column-wise wires, the M3D WLP could be a better design option than the M3D BLP.

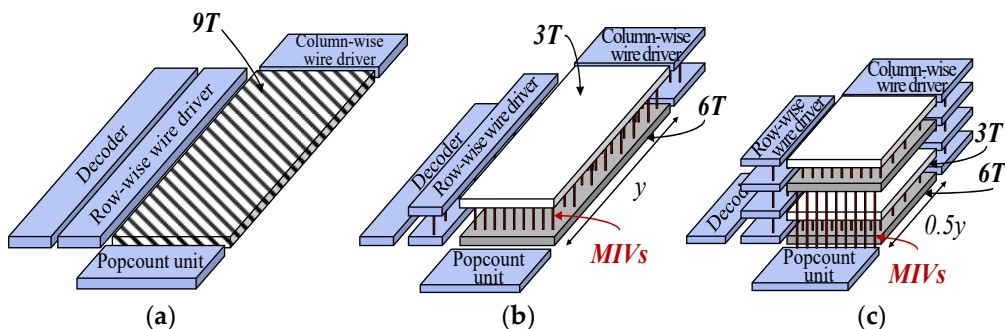

**Figure 2.** Three different CiM_SRAM designs. (**a**) CiM_SRAM (2D); (**b**) CiM_SRAM (M3D_2L); (**c**) CiM_SRAM (M3D_4L).

### 3.2. Overall Structure of BNN_Accel

Figure 3 depicts the overall structure of *BNN_Accel,* which consists of three blocks: an input buffer, *CiM_SRAM*, and global periphery logic. The input buffer stores the input feature maps (the input data of convolution layers) in 18 kB SRAM and provides the data to *CiM_SRAM* for the computation. Note 18 kB SRAM is large enough to store the input feature maps of each binarized convolution layer in commonly exploited BNNs (e.g., binarized VGG-16 [5–7]). *CiM_SRAM* is composed of 36 subarrays which store the kernel weights. With the input data provided from the input buffer and kernel weights stored in *CiM_SRAM*, all the *CiM_SRAM* subarrays perform XACs in parallel as explained in Section 3.1. Note, before the computations, the input feature maps and kernel weights are prepared in the input buffer and *CiM_SRAM*, respectively, based on the data mapping scheme which will be explained in Section 3.3. Then, the global periphery logic obtains the final output of the computations in the binarized convolution layer. The adder in the global periphery logic accumulates the partial popcount results from the *CiM_SRAM* subarrays to obtain the total popcount. The comparator then compares the total popcount with a predefined threshold and stores the final output to the buffer (i.e., when the total popount is less than the threshold, the final output is zero, otherwise, the final output is one). Lastly, the final output in the buffer is sent to the main memory.

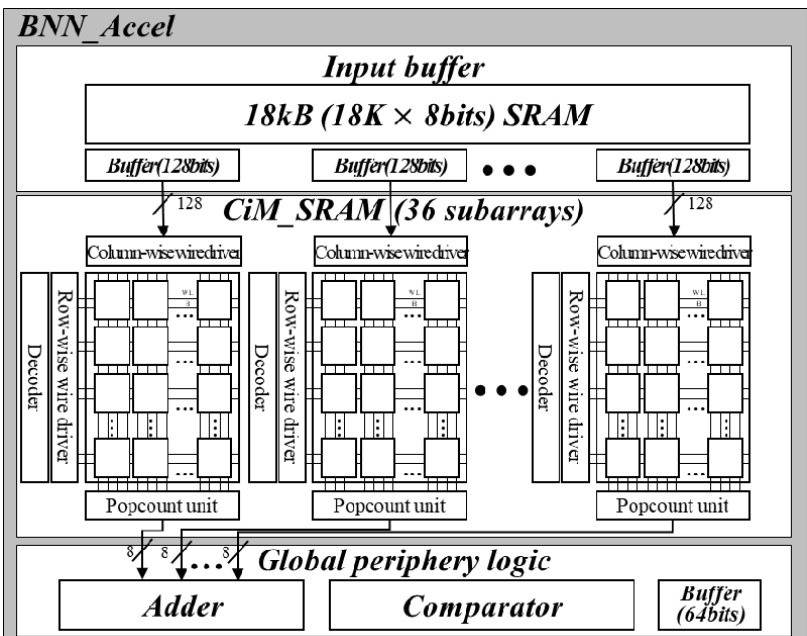

**Figure 3.** Overall structure of *BNN_Accel*.

### 3.3. Data Mapping for Parallel Computations in BNN_Accel

To efficiently perform parallel computations in *BNN_Accel,* we exploit a data mapping scheme for the input buffer and *CiM_SRAM* to store the input feature maps and kernel weights, respectively. We map the data for the parallel computation to a single row of a subarray. When the data size is too large to be stored in the single row, we map the rest of the data to a single row of another subarray. In other words, we do not map the data for the parallel computation to multirows of the same subarray. Figure 4 shows the data mapping example for the kernel weights in the 6th convolution layer of binarized VGG-16; there are 512 kernels with the size of $3 \times 3 \times 512$. Note, for each kernel, we simultaneously perform XACs. We map the data on each pixel (512 bits) of the nth kernel to the nth row (128 bits) of four subarrays. For example, we map the data on the pixel (1,1) of the first kernel to the first row of the 1st~4th subarrays, while we map the data on the pixel (1,2) of the first kernel to the first row of the 5th~8th subarrays. In this way, we map all the data on the same kernel to a single row of all the subarrays. Consequently, when all the *CiM_SRAM* subarrays operate in parallel, XACs for one kernel are performed at the same time.

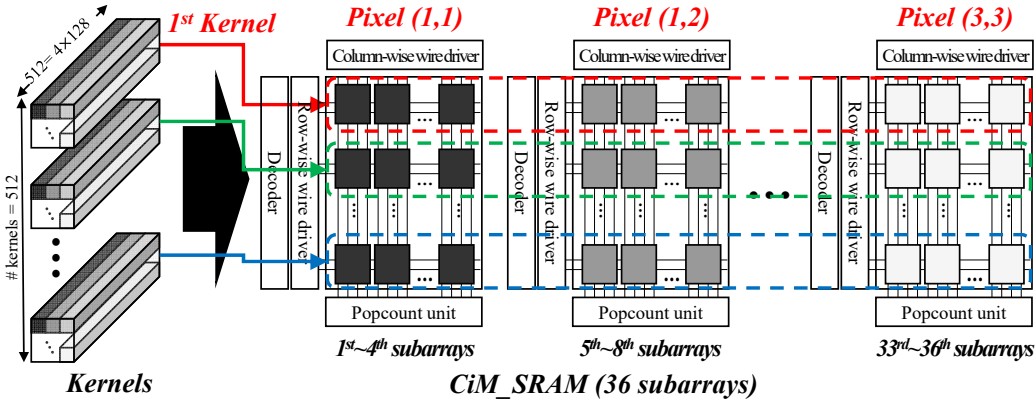

**Figure 4.** An example of data mapping scheme for parallel computations.

## 4. Evaluation

### 4.1. Evaluation Methodology

We modeled three different *CiM_SRAMs* with CACTI [11] to evaluate the latency and energy consumption. The size of each *CiM_SRAM* was 288 kB, since each *CiM_SRAM* was composed of 36 subarrays with 512-by-128 cell array as explained in Section 3. Note, the latency and energy of the cell were not shown in [9]. We modify CACTI to reflect the additional transistors and wires in *CiM_SRAM* as follows:

1.  We modeled the impact of additional transistors and wires on latency and energy in our modified CACTI, based on the Horowitz equation [11].
2.  We modeled the latency and energy of MIVs by adopting the MIV specification from [8].
3.  We modeled the area of *CiM_SRAMs* reflecting the cell structures; *CiM_SRAM (M3D_2L)* and *CiM_SRAM (M3D_4L)* had the same cell area as the conventional 6T SRAM, while *CiM_SRAM (2D)* had $1.5\times$ larger cell area than the 6T SRAM.

In addition, we implemented the Popcount unit with Verilog HDL. Then, we synthesized and placed-and-routed the Popcount unit with the Synopsys Design Compiler [12] and IC Compiler [13]. We exploited the Samsung System LSI 28 nm ASIC library. Meanwhile, since the 28 nm technology node does not exist in CACTI, we modeled *CiM_SRAM* based on the 22 nm technology node, which is the only sub-30 nm technology node in CACTI.

Table 1 shows the obtained latency and energy of the Popcount unit. Finally, we obtained the latency and energy of *CiM_SRAM* by adding the results of the Popcount unit to CACTI.

**Table 1.** Implementation Results.

|  | Popcount Unit | Input Buffer | Global Periphery Logic |
| --- | --- | --- | --- |
| Latency | 0.43 ns | 0.28 ns | 0.64 ns |
| Dynamic power | 2.77 mW | 35.88 mW | 3.07 mW |
| Leakage power | 2.21 μW | 29.32 μW | 2.63 μW |

Based on the three *CiM_SRAM* designs, we evaluated three different *BNN_Accels*: *BNN_Accel (2D)*, *BNN_Accel (M3D_2L)*, and *BNN_Accel (M3D_4L)*, which were *BNN_Accels* with *CiM_SRAM (2D)*, *CiM_SRAM (M3D_2L)*, and *CiM_SRAM (M3D_4L)*, respectively. We analyzed the execution time and energy consumption of each *BNN_Accel*, which executed the computations of each binarized convolution layer in binarized VGG-16 with Cifar-10 dataset [5]; in BNNs, the first convolution layer was not binarized to achieve high network accuracy [1]. We obtained the latency and energy of the input buffer and global periphery logic with the same process as the Popcount unit, as shown in Table 1. Then, we applied the number of cycles and energy taken by each block to gem5-aladdin [14]; *BNN_Accel* operated at 2.0 GHz.

In M3D integration, there was a concern about the thermal problem induced by high power density and low heat dissipation capability. Thus, we evaluated the peak temperature of each *BNN_Accel* with HotSpot 6.0 [15]. For a conservative thermal evaluation, each *BNN_Accel* was placed beside the big CPU core cluster consuming 4.0 W, which was the thermal design power (TDP) of the cluster [16]. Note the big CPU core cluster is the thermal hotspot in mobile SoCs [17]. To reflect the layer material property of M3D integration, we also set the thickness and thermal conductivity of the interlayer dielectric (ILD) to 100.0 nm and 1.4 W/m·K, respectively [8].

### 4.2. Evaluation Results

#### 4.2.1. Latency and Energy of *CiM_SRAM*

Figures 5 and 6 show the latency and energy consumption of each *CiM_SRAM*, when it performs XAC; note we present the access (read) latency of the conventional 6T SRAM in

Figure 5 for comparison with the XAC latency of *CiM_SRAMs.* The 6T SRAM has shorter latency than *CiM_SRAM (2D),* for the following reasons:

1. A read operation in the 6T SRAM uses a smaller number of transistors than an XAC operation in *CiM_SRAM (2D)*, which leads to shorter latency.
2. The 6T SRAM is composed of a smaller number of transistors/wires than *CiM_SRAM (2D),* which leads to lower parasitic capacitance.

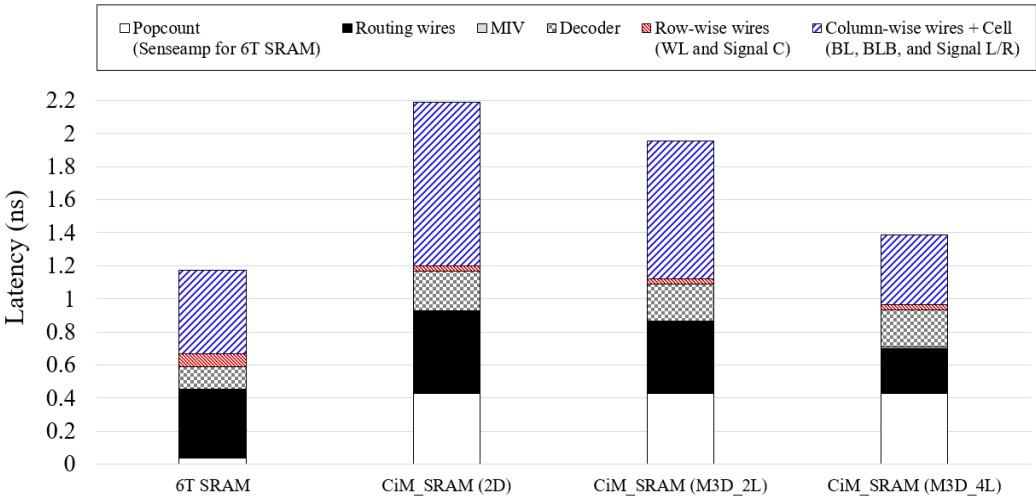

**Figure 5.** Latency of different SRAM designs (Note the latency value for 6T SRAM represents read latency and those for three different *CiM_SRAMs* are XAC operation latencies).

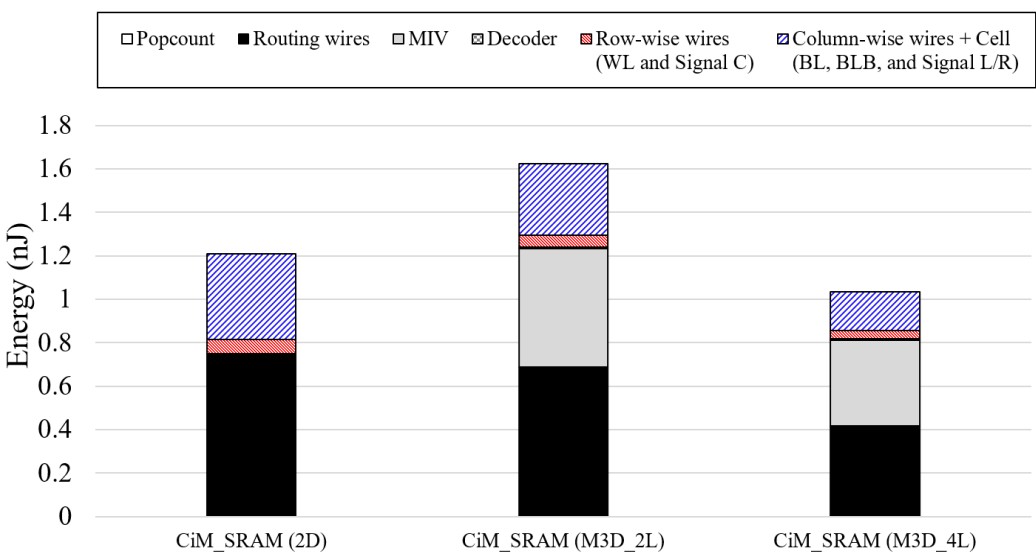

**Figure 6.** Energy consumption for different *CiM_SRAM* designs.

Though *CiM_SRAM (2D)* required much longer latency for an XAC operation than the read operation in the 6T SRAM, the latency could be reduced significantly by applying M3D BLP design to the *CiM_SRAM*. As shown in Figure 5, *CiM_SRAM (M3D_4L)* had 36.6% shorter latency than *CiM_SRAM (2D)* [9]. *CiM_SRAM (M3D_4L)* had shorter 2D wires (which included the column-wise wires, row-wise wires, and routing wires between subarrays) than *CiM_SRAM (2D),* resulting in shorter latency. The reduction of the 2D wire length was due to the following reasons:

1. Thanks to the transistor-level M3D integration, *CiM_SRAM (M3D_4L)* had 33.8% smaller cell area than *CiM_SRAM (2D).*

2. *CiM_SRAM (M3D_4L)* had shorter column-wise wires by adopting the M3D BLP.
3. Thanks to both (1) and (2), *CiM_SRAM (M3D_4L)* had smaller subarrays, which eventually reduced the length of the routing wires between subarrays.

In addition, *CiM_SRAM (M3D_4L)* reduced the latency by 10.7% compared to *CiM_SRAM (M3D_2L)* [9]. Though *CiM_SRAM (M3D_4L)* had the same cell area as *CiM_SRAM (M3D_2L)*, it led to shorter latencies of the routing wires between subarrays and column-wise wires, due to the M3D BLP; the latencies of *CiM_SRAM (2D), CiM_SRAM (M3D_2L),* and *CiM_SRAM (M3D_4L)* were 5 cycles, 4 cycles, and 3 cycles, respectively, at 2.0 GHz (the operating clock frequency of *BNN_Accels*).

As shown in Figure 6, *CiM_SRAM (M3D_4L)* consumed 14.0% less energy than *CiM_SRAM (2D).* The energy reduction also came from shorter 2D wires, thanks to the transistor-level M3D integration and M3D BLP. Though the energy of MIVs (i.e., $3 \times 512 \times 128$ MIVs in the cell array of each subarray) was non-negligible, the amount of energy reduction in 2D wires was higher than the additional MIV energy, attributed to shorter 2D wires. Additionally, *CiM_SRAM (M3D_4L)* reduced the energy by 36.4% compared to *CiM_SRAM (M3D_2L),* since it reduced the 2D wire energy thanks to the M3D BLP. Note, though *CiM_SRAM (M3D_2L)* reduced the 2D wire energy compared to *CiM_SRAM (2D),* *CiM_SRAM (M3D_2L)* consumed 34.3% more energy due to the MIV energy.

In Table 2, we summarize the results of *CiM_SRAMs* in terms of latency, energy, and area, which are normalized to those of *CiM_SRAM (2D)*; note the area indicated 288KB *CiM_SRAM* area. As described in Table 2, *CiM_SRAM (M3D_4L)* showed much better latency and energy than *CiM_SRAM (2D)*, with much smaller area, while *CiM_SRAM (M3D_2L)* showed worse energy efficiency due to its additional MIV energy.

**Table 2.** Normalized results of *CiM_SRAMs* in terms of latency, energy, and area.

|  | *CiM_SRAM (2D)* | *CiM_SRAM (M3D_2L)* | *CiM_SRAM (M3D_4L)* |
|---|---|---|---|
| Latency | 1.000 | 0.893 (10.7% lower) | 0.634 (36.6% lower) |
| Energy | 1.000 | 1.344 (34.4% higher) | 0.855 (14.5% lower) |
| Area | 1.000 | 0.711 (28.9% smaller) | 0.375 (62.5% smaller) |

4.2.2. Execution Time and Energy of *BNN_Accel*

Figure 7 shows the execution time and energy consumption of *BNN_Accels*, across the binarized convolution layers; note *BNN_Accel (2D)* is an BNN accelerator based not on 6T SRAM, but on *CiM_SRAM (2D)*. As shown in Figure 7a, *BNN_Accel (M3D_4L)* had a 39.9% shorter execution time than *BNN_Accel (2D)*, on average. *BNN_Accel (M3D_4L)* reduced the computation cycles for XAC by 40.0% (2 cycles), compared to *BNN_Accel (2D)*, resulting in shorter execution time. Additionally, *BNN_Accel (M3D_4L)* reduced the average execution time by 36.4%, compared to *BNN_Accel (M3D_2L)*. The reduction of the execution time also came from the reduced computation cycles for XAC. Note the computations of the binarized convolution layer consisted of many XACs.

As shown in Figure 7b, *BNN_Accel (M3D_4L)* consumed 23.2% less energy than *BNN_Accel (2D)*, on average. Since *BNN_Accel (M3D_4L)* reduced the energy for XAC as well as the execution time itself, it reduced the energy compared to *BNN_Accel (2D)*. Thanks to the same reasons, *BNN_Accel (M3D_4L)* reduced the average energy by 32.5%, compared to *BNN_Accel (M3D_2L)*.

In Table 3, we summarize the results of *BNN_Accels* in terms of execution time, energy and area, which are normalized to those of *BNN_Accel (2D)*; note the area includes popcount unit, input buffer, and global periphery logic. As described in Table 3, *BNN_Accel (M3D_4L)* outperformed *BNN_Accel (2D)*, in the perspective of execution time, energy, and area. Since *BNN_Accel (M3D_2L)* consumed more energy in *CiM_SRAM (M3D_2L)* (as shown in Figure 6), it resulted in slightly higher energy consumption then *BNN_Accel (2D)*. Consequently, combining transistor-level and subarray-level M3D integration improved performance and energy efficiency, with significant area reduction.

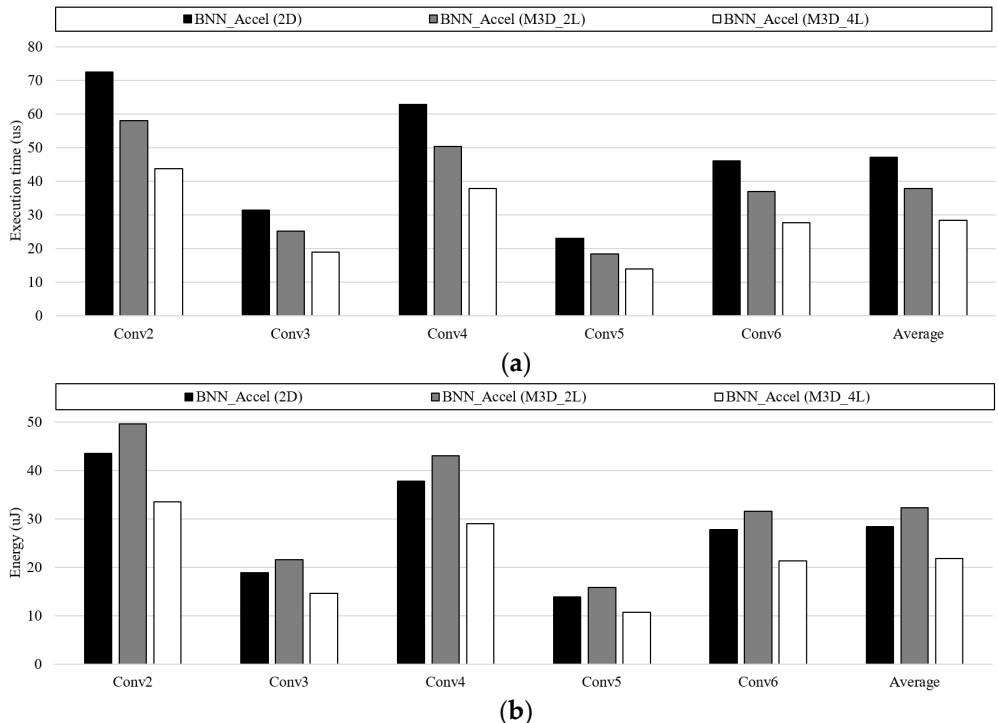

**Figure 7.** Execution time and energy consumption of three different *BNN_Accels* across the binarized convolution layers (i.e., Conv) (**a**) execution time results; (**b**) energy results.

**Table 3.** Normalized results of *BNN_Accel*s in terms of execution time, energy and area.

|  | *BNN_Accel (2D)* | *BNN_Accel (M3D_2L)* | *BNN_Accel (M3D_4L)* |
|---|---|---|---|
| Exec. time | 1.000 | 0.801 (19.9% faster) | 0.601 (39.9% faster) |
| Energy | 1.000 | 1.133 (13.3% higher) | 0.768 (23.2% lower) |
| Area | 1.000 | 0.759 (24.1% smaller) | 0.479 (53.1% smaller) |

### 4.2.3. Peak Temperature of *BNN_Accel*

We analyzed the peak temperature of each *BNN_Accel,* which was placed beside the thermal hotspot of the mobile SoC. The peak temperatures of *BNN_Accel (2D), BNN_Accel (M3D_2L),* and *BNN_Accel (M3D_4L)* were 71.8 °C, 76.9 °C, and 79.2 °C, respectively. Due to the high power density and low heat dissipation capability of M3D integration, the peak temperature of *BNN_Accel (M3D_2L)* and *BNN_Accel (M3D_4L)* was higher than that of *BNN_Accel (2D).* However, the peak temperature was still under the thermal threshold (80.0 °C in [18]), which does not invoke the thermal throttling.

### 4.3. Discussion on Monolithic 3D Fabrication Cost

According to our analysis, M3D_4L is expected to provide much better performance, higher energy efficiency, and even significant area reduction, compared to the conventional 2D design. However, M3D integration is not a mature technology, we need to consider additional fabrication costs compared to the conventional 2D fabrication; for example, some additional design steps are required for stacking multiple layers such as increasing the number of masks. In addition, to prevent damage to the bottom layer, M3D requires low temperature fabrication for each stacked layer, which requires additional equipment for low temperature fabrication (e.g., laser annealing, etc.). To reduce the design costs of M3D fabrication, many researchers have studied M3D across various levels. For example, Or-Bach et al proposed an M3D fabrication method that could be compatible with the conventional 2D process [19]. Additionally, Jiang et al. presented the use of 2D materials for M3D integration [20], which would lead to significant cost reduction in M3D integration,

even comparable to 2D ICs. Based on such studies, M3D is already considered to be more cost-effective than the conventional TSV-3D for logic fabrication. Consequently, various studies will enable M3D to be applied to commercial products in the near future.

## 5. Conclusions

In this paper, we proposed the *BNN_Accel* for the binarized convolution layers of BNNs, which consisted of *CiM_SRAM*, input buffer, and global periphery logic. We adopted the subarray-level M3D integration (the M3D BLP) to *CiM_SRAM (M3D_4L)* for the wire length reduction, which reduced the latency and energy, compared to the *CiM_SRAM (2D)*. Across the binarized convolution layers in a BNN, our system-level evaluation shows that the *BNN_Accel (M3D_4L)* reduces the average execution time and energy by 39.9% and 23.2%, respectively, compared to *BNN_Accel (2D)*. Though we only showed the results using the binarized VGG-16, our proposed *BNN_Accel* could provide energy efficiency for more complex networks, by adjusting the size of input buffers and *CiM_SRAM*s, considering the size of inputs, size of kernels, etc.

**Author Contributions:** J.H.C.; methodology, data analysis, writing—original draft preparation, Y.-H.G.; data analysis, validation, writing—review and editing, S.W.C.; data analysis, validation, writing—review and editing, supervision. All authors have read and agreed to the published version of the manuscript.

**Funding:** This work was supported by the National Research Foundation of Korea (NRF) grant funded by the Korea government (MSIT) (No. 2020R1A2C2003500) and funded by the Ministry of Science and ICT for Original Technology Program (No. 2020M3F3A2A01082329). This work also has been conducted by the Research Grant of Kwangwoon University in 2020.

**Data Availability Statement:** The data presented in this study are available on request from the corresponding author.

**Conflicts of Interest:** The authors declare no conflict of interest.

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
