# Peer review of "A System-Level Exploration of Binary Neural Network Accelerators with Monolithic 3D Based Compute-in-Memory SRAM"

_electronics, doi:10.3390/electronics10050623_

Round 1

Reviewer 1 Report

The authors investigate the use of the 3D design of SRAM memory cells to perform in-memory computation of Binary Neural Networks (BNNs). The study compares two different designs to the standard planar - 6T cells in terms of energy efficiency, latency, memory. The results of the proposed M3D_4L are interesting, demonstrating good potentials in terms of energy and area reduction. However, some improvements are possible to increase the quality of the work. -) I suggest summarizing the results discussed in Section 4.2 by using a table to compare the different technologies in terms of area, energy per inference, latency to make it more clear to the reader. -) Figure 5 shows that XAC operations latency for the standard 6T-cell is reduced compared to the one for 3D technology. However, Figure 7 shows that for an entire layer, 6T technology seems to be slower. Authors should better clarify how this reduction in latency is possible given the slower basic operation (for instance, to demonstrate how the number of XAC operations changes for a given layer depending on the technology) -) It would be interesting to compare the M3D_4L technology to some state of the art solutions in terms of energy efficiency, latency and area, if possible. -) What are the drawbacks of the proposed M3D_4L technology in terms of design steps compared to the standard 6T technology? A discussion (even a brief summary) would be interesting in this respect.

Author Response

We'd like to thank the reviewer for helpful comments.

Please see the attached response letter, which includes our response to all the reviewers' comments. 

Reviewer 2 Report

Dear authors,

I read with interest your manuscript and I feel satisfied with the present version of the text.

The claims made in the abstract are justified, the methodology is sound.

Author Response

(The authors gave the same response as above.)

Reviewer 3 Report

This manuscript presents a Binary Neural Network (BNN) accelerator, which include a Compute-in-Memory(CiM) SRAM that have been developed by the authors by applying subarray-level M3D integration techniques. The paper is well-written and the authors clearly explain the work done. I am not an expert in this specific field, but I think that the authors have justified properly the novelty of their approach using very recent references. In fact, the authors assert that they present the first study that investigates the system-level impacts of subarray-level M3D integration in the CiM SRAM on BNN accelerators. In addition, they have made an exhaustive comparison between the proposed CiM SRAM (called M3D_4L) and other state-of-the-art implementations. M3D_4L obtains significance improvements in terms of latency and energy consumption. The resulting BNN accelerator also achieves advantages in terms of execution time and energy.

The authors say that 18KB SRAM is enough to store the input feature maps of each binarized convolution layer of a VGG-16. I think that it would be a good idea to include a discussion about the necessary changes to implement more complex convolutional neural network models like VGG-19.

I suggest the authors to indicate explicitly in Figure 1 that only part of the figure is reproduced from [6] because XNOR operation is not included in that reference.

Author Response

(The authors gave the same response as above.)

Round 2

Reviewer 1 Report

The authors improved the quality of the manuscripts according to the most of the suggestions provided.

The manuscript can be accepted in this form.